# Lack of Consistent Association between Asthma, Allergic Diseases, and Intestinal Helminth Infection in School-Aged Children in the Province of Bengo, Angola

**DOI:** 10.3390/ijerph18116156

**Published:** 2021-06-07

**Authors:** Margarete Arrais, Ofélia Lulua, Francisca Quifica, José Rosado-Pinto, Jorge M. R. Gama, Philip J. Cooper, Luis Taborda-Barata, Miguel Brito

**Affiliations:** 1Department of Pulmonology, Military Hospital, Luanda 12195, Angola; mararrais@hotmail.com (M.A.); ofelialulua@hotmail.com (O.L.); adeliaquifica@yahoo.com.br (F.Q.); 2CISA—Centro de Investigação em Saúde de Angola/Health Research Center of Angola, Caxito 9999, Angola; Miguel.brito@estesl.ipl.pt; 3Department of Immunoallergology, Hospital da Luz, 1500-650 Lisbon, Portugal; rosadopinto@mail.telepac.pt; 4Center of Mathematics and Applications, Faculty of Sciences, University of Beira Interior, 6201-001 Covilhã, Portugal; jgama@ubi.pt; 5Institute of Infection and Immunity, St George’s University of London, London SW17 0RE, UK; pcooper@sgul.ac.uk; 6School of Medicine, International University of Ecuador, 170411 Quito, Ecuador; 7Department of Immunoallergology, Cova da Beira University Hospital Centre, 6200-001 Covilhã, Portugal; 8CICS—Centro de Investigação em Ciências da Saúde/Health Sciences Research Centre, University of Beira Interior, 6200-506 Covilhã, Portugal; 9Health and Technology Research Center (H&TRC), The Lisbon Higher School of Health Technology, The Polytechnic Institute of Lisbon, 1990-096 Lisbon, Portugal

**Keywords:** Angola, asthma, atopy, children, eczema, helminths, rhinoconjunctivitis

## Abstract

Epidemiological studies have shown conflicting findings on the relationship between asthma, atopy, and intestinal helminth infections. There are no such studies from Angola; therefore, we aimed to evaluate the relationship between asthma, allergic diseases, atopy, and intestinal helminth infection in Angolan schoolchildren. We performed a cross-sectional study of schoolchildren between September and November 2017. Five schools (three urban, two rural) were randomly selected. Asthma, rhinoconjunctivitis, and eczema were defined by appropriate symptoms in the previous 12 months: atopy was defined by positive skin prick tests (SPT) or aeroallergen-specific IgE; intestinal helminths were detected by faecal sample microscopy. In total, 1023 children were evaluated (48.4% female; 57.6% aged 10–14 years; 60.5% urban). Asthma, rhinoconjunctivitis, or eczema were present in 9%, 6%, and 16% of the studies children, respectively. Only 8% of children had positive SPT, but 64% had positive sIgE. Additionally, 40% were infected with any intestinal helminth (*A. lumbricoides* 25.9%, *T. trichiura* 7.6%, and *H. nana* 6.3%). There were no consistent associations between intestinal helminth infections and asthma, allergic diseases, or atopy, except for *A. lumbricoides,* which was inversely associated with rhinoconjuctivitis and directly associated with aeroallergen-specific IgE. We concluded that, overall, intestinal helminth infections were not consistently associated with allergic symptoms or atopy. Future, preferably longitudinal, studies should collect more detailed information on helminth infections as part of clusters of environmental determinants of allergies.

## 1. Introduction

Epidemiological studies have demonstrated a worldwide increase in the prevalence of allergic diseases such as asthma, rhinitis, and eczema, especially in children [1]. The International study of Asthma and Allergies in Childhood (ISAAC), done in many regions worldwide including in Africa, showed an increase in the prevalence of these diseases over a 5–10-year period [2].

The increase in allergic disease prevalence has been attributed to various factors including changes in environmental, socioeconomic, and lifestyle factors, as well as exposures to childhood infections including intestinal helminth parasites. It is estimated that more than two billion people are infected with intestinal helminths worldwide, especially in Africa, Latin America, China, and East Asia, where they are associated with poor hygiene and sanitation, and lack of access to health services [3].

Helminth parasites are potent modulators of the host allergic inflammatory response, via mechanisms such as induction of IL-10 and TGF-β in peripheral blood mononuclear cells (possibly regulatory T cells) and regulatory B cells [4,5], and it has been suggested that helminth infections including intestinal helminths may reduce allergic disease prevalence in populations where these infections are endemic [6]. However, although infections with *Ascaris lumbricoides*, *Trichuris trichiura*, and hookworm have been associated with decreased prevalence of atopy measured by allergen SPT in two systematic reviews of epidemiological studies [7,8], their effects on allergic diseases have shown more discrepant results [6,7,8].

The prevalence of asthma and allergic diseases appears to be increasing in African children [9], while human helminth infections remain a major public health problem [10,11]. Several studies addressing the relationship between intestinal helminths and allergy have been done in African countries showing inverse, positive, or no associations [7,8,12,13,14,15,16,17,18,19,20,21,22,23]. However, there are no previous studies addressing the relationship between intestinal helminth infections and allergy from Angola. Thus, the aim of the present study was to evaluate the relationship between the presence of helminth infections and prevalence of atopy, asthma and other allergic diseases among children living in the Angolan province of Bengo.

## 2. Materials and Methods

### 2.1. Study Area and Population

The study was conducted in Angola, province of Bengo, municipality of Dande, Communes of the Caxito urban region and Úcua rural region in the Dande Health Demographic Surveillance System (HDSS-Dande) area. The HDSS from the Health Research Center of Angola (CISA) covers an area of 4700 km², corresponding to 69 neighbourhoods, about 16,000 houses, and a population estimated to be 60,000 inhabitants [24]. The province of Bengo is located 60 km from the province of Luanda, has an extension of 33,016 km², and a population estimated at 356,641 inhabitants. The municipality of Dande is the most densely inhabited, in which 62% of the population of the province is found. The province is predominantly rural with a tropical, semi-arid climate and annual temperatures ranging from 22 to 32 °C.

### 2.2. Study Design

This cross-sectional study of children was done between September and November 2017. Of the 33 schools in the study area, 5 (15%) were randomly selected, of which 3 were in urban and 2 in rural areas.

### 2.3. Sample

All children aged 5 to 14 years were invited to participate. Any child with symptoms of possible infectious lung disease, or other chronic infectious diseases were excluded after clinical examination on the day of data collection. Children whose parents or guardians did not agree to participate in the study were also excluded.

The sample size to estimate a prevalence of at least 13% with 3% precision was calculated using OpenEpi (https://www.openepi.com/; accessed on 30 May 2017) from a population of 17,873 children aged 5–14 years in CISA HDSS-Dande. Estimates of prevalence were based on previous studies from Bengo, Luanda or elsewhere in Africa: intestinal helminths 32% [25], asthma 13% [26], allergic rhinitis 19% [27], atopic eczema 20% [26], and atopy 25% [28]. The estimated sample size of 471 was increased to 1000 based on a ~50% response rate.

### 2.4. Questionnaires

Data collection was carried out using ISAAC methodology [29,30]. Questionnaires were administered to parents or guardians in the presence of the child to collect information on symptoms of asthma, rhinitis, and eczema, as well as demographic and environmental data. Questionnaires were translated into and validated in Portuguese [29,31,32], and were administered by a team of trained healthcare workers using standardised methodology.

### 2.5. Allergen Skin Prick Tests

Skin prick tests were performed by two experienced nurses, supervised by a pulmonologist. Tests were carried out in accordance with the standardised procedure of the European Academy for Allergology and Clinical Immunology [33]. Twelve aeroallergens were used (*Dermatophagoides pteronyssinus*, *Dermatophagoides farinae*, *Blomia tropicalis*, Grass pollen mix, weed mix, *Aspergillus* species, *Cladosporium* species, *Mucor mucedo*, *Alternaria alternata*, cockroach mix, and dog and cat epithelia). Histamine was used as the positive control and saline as negative control (Leti Laboratories, Tres Cantos, Madrid, Spain). A small drop of each allergen extract was placed on the anterior surface of a forearm and pricked with a lancet (Leti Laboratories, Tres Cantos, Madrid, Spain). Weals were measured after 15 min and were considered positive if >3 mm (in the absence of a reaction to saline and a positive reaction to histamine) [33].

### 2.6. Inhalant Phadiatop Screening Test

Five millilitres of blood was collected and the serum was separated. Sera were aliquoted and stored at −20 °C prior to analysis using the inhalant Phadiatop^TM^ test (Immunocap 250^®^, ThermoFisher Scientific, Uppsala, Sweden), which is a fluorescence enzyme immunoassay that detects specific IgE to 11 common inhalant aeroallergens [34,35]. A positive screening test for aeroallergen-specific IgE (sIgE) was defined by Phadiatop^TM^ values ≥ 0.35 kU_A_/L.

### 2.7. Stool Parasitological Examination

A single stool sample was obtained from each child and processed using the Parasitrap^®^ concentration (Parasitrap fixsepar Eco System, VWR International GmbH, Darmstadt, Germany) and Kato–Katz methods (Sterlitech Kit Katokatz method, Kent, Washington, DC, USA) to detect the presence of eggs and larvae of intestinal helminths and *Schistosoma haematobium*. Slides were examined independently by two technicians within 2 h of preparation following WHO recommendations [36]. The presence of any characteristic eggs or larvae using either method was considered positive. Intestinal helminth infection intensities were estimated using results of the Kato–Katz method and classified as light, moderate, or heavy according to WHO recommendations [37].

### 2.8. Definitions

Asthma was defined by the presence of wheezing or whistling episodes in the chest in the previous 12 months; rhinitis was defined by the presence of sneezing, nasal secretions, or congestion, not associated with a cold or influenza in the previous 12 months; rhinoconjunctivitis was defined by the presence of nasal symptoms and itching and tearing of the eyes in the previous 12 months; and eczema was defined by the presence of an itchy rash in the skin that waxed and waned in the previous 12 months, in accordance with ISAAC [29,31]. Atopy was defined either as a positive skin prick test to any aeroallergen [33] or a positive test for sIgE [34,35]. Eosinophilia was defined as an elevation of peripheral blood eosinophil counts above 0.5 × 10^9^/L [38].

### 2.9. Statistical Analysis

Categorical variables were described using frequencies and percentages, and quantitative variables using mean, median, standard deviation, and maximum and minimum levels. Associations between variables of interest and potential risk factors were evaluated using logistic regression with adjustment for potential confounders selected on the basis of *p* < 0.2 in univariate analyses. Mann–Whitney U test was used for comparison of quantitative variables between two groups. Chi-square test was used to compare frequencies of categorical variables. Statistical significance was inferred by *p* < 0.05. Data were analysed using SPSS (version 25, IBM, Armonk, NY, USA).

### 2.10. Ethical Considerations

This study protocol was approved by the Ethics Committees of the Ministry of Health of Angola (Date of approval: 2 May 2017) and the University of Beira Interior, Portugal (Protocol code: CE-UBI-Pj-2017-024; date of approval: 10 June 2017). Schools were included after seeking approval of school headmasters. Parents and guardians gave written informed consent for their children to participate. Antiparasitic treatment (albendazole and/or praziquantel, as appropriate) was provided for free to all children with positive stool samples. Children with asthma, rhinoconjunctivitis, and atopic eczema requiring treatment were referred to the health system and advice on allergen avoidance provided to parents of children with positive skin prick tests.

## 3. Results

### 3.1. Sociodemographic Data

The sample consisted of 1023 children aged 5 to 14 years from five schools. Sociodemographic characteristics of the sample are shown in Table 1. 

Approximately half the children were boys (51.6%), and most children were aged 10 to 14 years (57.6%) and lived in an urban area (60.5%). Most mothers had 4 years or less of schooling (71.4%; data not shown).

### 3.2. Asthma and Allergic Diseases

Asthma, rhinoconjunctivitis, and eczema symptoms were reported for 9.3%, 6.0%, and 15.9% of children, respectively. Associations between sociodemographic factors (including helminth infections) and allergic diseases symptoms are shown in Table 2 and Table 3.

Both in univariate (Table 2) and in multivariable analyses (Table 3), the prevalence of rhinoconjunctivitis was significantly greater in older compared to younger children, and eczema was more frequent in girls and children living in urban areas.

### 3.3. Atopy

Positive skin prick tests were observed in 8.0% of children and the most frequent sensitisations were to mites (67.2% of positive children), cockroaches (16.1%), and fungi (8.8%). No statistically significant associations were observed between SPT prevalence and age, sex, and area of residence (Table 4).

Of the 1023 children evaluated, data on the presence of sIgE were available for 964 of whom 618 (64.1%) had positive tests. sIgE was significantly more frequently observed in boys compared to girls, as well as in rural compared to urban children (Table 4).

### 3.4. Intestinal Helminth Infection

A total of 407 (39.8%) children were infected with any intestinal helminth, with *A. lumbricoides* (25.9%); *T. trichiura* (7.6%), and *Hymenolepis nana* (6.3%) being most frequent (Table 1). *A. lumbricoides* infections were more frequent in younger compared to older children (OR 1.77, 95% CI 1.33–2.34, *p* < 0001). Infections with both *A. lumbricoides* (OR 1.35, 95% CI 1.02–1.80, *p* = 0.037) and *T. trichiura* were more frequent in children living in a rural area (OR 1.78, 95% CI 1.12–2.82, *p* = 0.015). Information on previous anthelmintic treatments was missing for 70 children (Table 1), and they were not analysed further. In comparison with non-infected children, those infected with intestinal helminths had more frequent peripheral blood eosinophilia (82.4% (299/364) vs. 62.3% (408/655); OR = 1.411, 95% CI 1.071–1.860, *p* = 0.014), and higher eosinophil counts (median 9.4% (IQR 11.2%) vs 7.8% (9.5%), *p* = 0.011). The total serum IgE levels measured in 964 children were also significantly higher in infected (*n* = 387) than in non-infected children (*n* = 577) (median 834.90 IU/L (IQR = 1896.60) vs. median 583.80 (IQR = 1605.5), *p* < 0.001).

### 3.5. Relationship between Intestinal Infections and Allergic Diseases and Atopy

No significant associations were observed between intestinal helminth infections and parasite burdens and symptoms of allergic diseases, except for an inverse association between rhinoconjunctivitis and infection with *A. lumbricoides*, as detected by univariate (Table 2) and multivariable (Table 3) analyses, with some evidence for significantly stronger inverse effects on rhinoconjunctivitis with the high parasite burdens of *A. lumbricoides* (Table 2).

Globally, no significant associations were observed between intestinal helminth infections and markers of atopy (SPT and sIgE) except for *A. lumbricoides*, which was significantly positively associated with sIgE (Table 4).

## 4. Discussion

To our knowledge, this is the first epidemiological study from Angola to analyse the association between intestinal helminth infections and allergy in children. To address this question, we recruited and sampled a representative group of schoolchildren within a demographic surveillance system in the Province of Bengo, known to be endemic for intestinal helminth parasites [25,39]. *A. lumbricoides* infections, particularly at high parasite burdens, were inversely associated with symptoms of rhinoconjunctivitis, but positively associated with atopy detected by sIgE. No other intestinal helminths were associated with allergic diseases or atopy, measured by either SPT or sIgE.

The first issue that should be highlighted concerns the fact that we mainly found a non-consistent association between intestinal helminth infection and atopy and/or allergic diseases. In this context, our study showed a prevalence of asthma of 9.3%, rhinitis of 21.6%, and eczema of 15.9%, values that are slightly lower than those in children (15.7%, 19.0%, and 18.4%, respectively) [27] and adolescents (13.4%, 26.9%, and 20.2%, respectively) [26] in the province of Luanda. In addition, the prevalence of positive SPT in our study was quite low (8%) in contrast to what has been observed among children from other urban and rural areas in Africa (reviewed by Mpairwe) [40], although it was similar to that seen in 3-year-old Ethiopian children (8.7%) [19]. Our observations of a low prevalence of SPT and a prevalence of asthma and allergic diseases lower than previously observed in Luanda, in a region of high prevalence of intestinal helminth infections (Bengo), raised the question of whether such infections might affect the prevalence of atopy and allergic diseases in children from that area. However, we only found a significant protective effect of *A. lumbricoides* infection against symptoms of rhinoconjunctivitis, but not asthma or eczema, with some evidence that stronger protective effects with higher parasite burdens. We did not observe significant associations between other types of helminth parasite and allergic outcomes.

Previous studies of associations between helminths and allergies done elsewhere in Africa in populations with comparable sociodemographic and environmental characteristics, and with sample sizes of 300 or more children, have shown contrasting results with protective effects [12,13,14,15], positive associations [15,16,17], and no effects [18,19,20,21,22,23]. Four reports are used here as examples of a lack of consistent positive or negative association. A study done in Ghana among 1482 6–15-year-old children from rural and urban areas did not observe associations between atopy (detected using SPT with house dust mites, cockroaches, and peanuts, and prick-prick tests using fresh fruits) and children infected with hookworm, *A. lumbricoides,* or *T. trichiura* [18]. However, the prevalence of intestinal helminth infection was low in this study—hookworm—6.76%, *Trichuris*—1.37%, and *Ascaris*—7.06%—and this may have affected the power of the study. In another study done in Ghana in 1385 5–16-year-old children from rural and urban settings, again no associations were observed between helminth infections (*A. lumbricoides*, *T. trichiura*, and hookworm) and wheeze/asthma or atopy (detected by positive SPT to mites, cockroaches, or peanuts), although *Schistosoma* spp. infection was associated with a reduced prevalence of SPT, and *T. trichiura* was associated with an increased prevalence of positive cockroach but not mite SPT [12]. The prevalence of helminth infection was higher in this study: any helminth—23.1%, hookworm—9.9%, *Ascaris*—6.2%, *Trichuris*—1.9%, and *Schistosoma* spp.—9.5%. In a study from Ethiopia involving 878 children aged 3 years living in rural and urban settings, no associations were observed between intestinal helminth infection (any helminth—8.5%; hookworm—*4.9%, A. lumbricoides*—4.3%, and *T. trichiura*—0.1%) and atopy (measured using SPT to *Der. pteronyssinus* and cockroach) or symptoms of wheezing, eczema, and hay fever [19]. Finally, in a study done in Rwanda in 3041 children aged 7–14 years living in rural and urban areas, no associations were observed between intestinal helminth infections (any helminth—23.1%, predominant infections: *Ancylostoma*—9.8% and *T. trichiura*—5.6%) and risk of vernal keratoconjunctivitis [20].

In contrast, a few studies have shown protective associations between intestinal helminth infection and allergy in African settings. Two examples: a nested case-control study in 773 black South African children aged 8–12 years from rural and urban areas with a high prevalence of intestinal helminths (61% for *A. lumbricoides* and 33% for *T. trichiur*a) showed that *A. lumbricoides* was associated inversely with atopy (SPT using *Der pteronyssinus*, *Der farinae*, *Blomia tropicalis*, cockroach, two grasses, three fungi, and cat and dog dander), but positively associated with atopy-related exercise-induced bronchospasm [15]. Additionally, a study of 7155 rural and urban Ethiopian children aged 1–4 years in a region with a high prevalence of intestinal helminths (*Trichiuris*—45%; *Ascaris*—38%; *Ancylostoma*—10%) showed an inverse association between *A. lumbricoides* infection, particularly at high parasite burdens, and risk of having wheeze [17].

Finally, a few studies have shown that helminth infections may be associated with increased atopy (detected by SPT) and/or increased prevalence of asthma or other allergic disease. Two examples: a nested case-control study designed to assess early life risk factors for atopic dermatitis, performed in 306 1–5-year-old Ethiopian children from a region with relatively high prevalence of intestinal helminth infection (*A. lumbricoides*—28%; *T. trichiura*—26%; hookworm—4%) showed the risk of having atopic dermatitis (as assessed by self-reported, ISAAC-based criteria) was significantly higher in children infected with *T. trichiuria* [17]. Another, cross-sectional, study was performed in several households from fishing villages on Lake Victoria islands, in Uganda, in an area with a high prevalence of *Schistosoma mansoni* (51%) and intestinal helminth infections (hookworm—22%; *S. stercoralis*—12%; *T. trichiura*—10%; *A. lumbricoides* infection was low—1%) and included 2316 individuals with a broader age range than in our study (mean = 24; median = 8 IQR = 8.32 years). Among various other factors, the same study also analysed the influence of helminths (*Necator americanus*, *Stronglyloides stercoralis*, *T. trichiura*, *Mansonella perstans*, *A. lumbricoides*, and *S. mansoni*) on the percentage of atopic individuals (19%, detected by SPT with mites—*Dermatophagoides* mix and *Blomia tropicalis,* and German cockroach) and also on manifestations of reported wheeze [16]. Although the other intestinal helminths and *S. mansoni* did not affect SPT, *T. trichiura* was associated with an increased risk of having positive SPT to any allergen, the house dust mite *Der pteronyssinus,* and cockroaches, and *A. lumbricoides* was associated with an increased risk of having wheeze in individuals >5 years.

Differences in findings between studies may be attributed to a variety of factors such as genetic background of the populations, sociodemographic aspects that may be confounding factors, various methodological aspects such as study power-related issues, different definitions of atopy and atopic disease outcomes, methods of parasite detection, predominant species of intestinal helminth parasites, intestinal helminth endemicity, parasite burden, the severity and frequency of intestinal helminth infection, the time when the infection occurred (recent or late), or the effect of previous and/or current treatments. When results of different studies were pooled in a meta-analysis done in 2011, intestinal helminths were more frequently inversely associated with atopy (measured using SPT), although the effects or magnitude of the effects varied by parasite type, being more reproducibly observed with *A. lumbricoides* and *T. trichuria* with a general effect on SPT positivity rather than on specific aeroallergens [8]. Nevertheless, as far as the impact of intestinal helminth infections on asthma or wheeze is concerned, results tend to show less consistent data, with a meta-analysis of 33 observational studies up to 2006, mostly from outside Africa, concluding that intestinal helminth infections were not protective of asthma, although hookworm may be associated with a reduced risk and *A. lumbricoides* with an increased risk of having asthma symptoms [7]. In addition, differences in results across different studies may also have to do with varying percentages of children who had previously (and how long before) received anti-helminth treatments. This is relevant because at least some anti-helminth treatments have been shown, in a murine model, to attenuate T2-type allergic inflammation in the lung [41]. In any case, in our study, results were not different between previously helminth-treated and -untreated children (data not shown).

Finally, it should be highlighted that discrepant results in terms of the relationship between helminth infections and atopy or asthma have also been reported and analysed in in various studies, which are reflected in meta-analyses including data from countries outside Africa, suggesting that the discrepancies are not geographically restricted [7,8]

The second issue that should be addressed concerns the discrepancy in results between SPT and allergen-specific IgE screening test (Phadiatop^TM^) in our study and their differential associations with intestinal helminth infection. Most studies analysing the relationship between intestinal helminth infections and atopy have used SPT as a marker of atopy, particularly in low-income countries. In fact, in such countries, serum tests that detect allergen-specific IgE may be significantly more frequently positive than the equivalent SPT and less frequently associated with allergic diseases [42]. Measurement of serum allergen-specific IgE as a marker of atopy may be unreliable in the context of endemic intestinal helminth infections because of cross-reactive carbohydrate determinants (CCDs) such as glycans, as was demonstrated in a recent study [43]. In this study, a high proportion of Ecuadorian children living in a highly endemic region for intestinal helminths (50% of children had *Ascaris* and *Trichiuris* in their stools) had elevated serum levels of IgE to α-Gal, a glycan (CCD), and, furthermore, such positive IgE levels to this CCD were associated with exposure to *A. lumbricoides*. In fact, the frequency of α-Gal sensitisation, independently of the levels of α-Gal-specific IgE, was clearly associated with the level of *Ascaris*-specific IgE. It is thus possible that Phadiatop^TM^ may be picking up this cross-sensitisation, thereby artificially becoming positive in a very high proportion of children living in endemic areas for intestinal helminths. This possibility was backed up by three studies carried out in Africa. The first two related studies, done in urban and rural Ghanaian schoolchildren, showed that anti-glycan IgE responses may be dominant over those of common allergen-specific IgE, namely peanuts [44], and preferentially associated with *Schistosoma* infection [22]. The third study, performed in blood collected from urban and rural Ugandan schoolchildren, showed that aeroallergen-specific IgE tests were more frequently positive than SPT to the same allergens, and also that such serum IgE tests were artificially positive to the tested allergens (*Der pteronyssinus*, German cockroaches, and peanuts), since a high proportion of this positivity was attributable to CCD-specific IgE and not to true atopy [45]. Furthermore, in vitro inhibition of IgE binding to CCDs has been shown to increase diagnostic selectivity of allergen-specific tests [46]. We thus believe that the clear discrepancy between SPT and serum aeroallergen-specific IgE tests in our study may have been due to CCD-specific IgE, which makes it difficult to interpret the results when atopy is based on specific IgE tests.

Some limitations may have influenced our results. First of all, this is a study based on the responses of children, parents, or caregivers of children, especially those living in rural areas, who had some limitations in understanding and interpretation of the questions of the ISAAC questionnaire, which may have led to various biases. Secondly, although we surpassed the minimal required sample size, we believe that the relatively small sample size of our study may also have interfered with our ability to detect significant associations. Thirdly, although in our study used one of the broadest batteries of aeroallergens for SPT, the battery was not validated for use in Angola, and may have missed cases related to pollen allergies. In addition, our results showed a prevalence of intestinal helminth infection of 35.7% in school-aged children, which is relatively similar to that found in a study published in 2012 [47], but lower than the national average of 40% [48]. In any case, the prevalence of intestinal helminth infections in our study is moderately high, as documented in various studies conducted in African countries [49,50,51,52], namely in Angola [25,39,47]. However, is important to bear in mind that our results may be underestimated, since we harvested only one stool sample from each child, whereas it has been described that diagnostic accuracy can be higher if more than one sample is analysed [53]. Finally, we did not address some confounding factors such as family history of allergic diseases and atopy, other infections such as tuberculosis, which has a high prevalence in Angola, and only had basic information on previous parasitological history, antiparasitic treatments, and socioeconomic and environmental factors, which are relevant in the analysis of risk factors. Nevertheless, we followed a thorough and validated approach regarding diagnosis of atopy, allergic diseases, and helminth infection, and this study produces novel information regarding these aspects in Angola.

## 5. Conclusions

In conclusion, we observed an inverse association between infection with *A. lumbricoides* and symptoms of rhinoconjunctivitis among schoolchildren living in Angola, but we did not observe any other significant associations for any other parasite types with SPT and allergic diseases. Further studies with larger samples from different regions of Angola with varying prevalence of intestinal helminths and more complete information on relevant environmental exposures are needed to better understand this relationship.

## Figures and Tables

**Table 1 ijerph-18-06156-t001:** Sociodemographic characteristics of study sample of 1023 children.

Parameter	*N* (%)
Sex	
Female	495 (48.4)
Male	528 (51.6)
Age range (year)	
5–9	434 (42.4)
10–14	589 (57.6)
Residence	
Urban	619 (60.5)
Rural	404 (39.5)
Anthelmintic treatment	
Yes	392 (38.3)
No	561 (54.8)
No data	70 (6.8)
Eosinophilia *	
Yes	707 (69.1)
No	316 (30.9)
Allergic diseases	
Asthma	95 (9.3)
Rhinoconjunctivitis	61 (6.0)
Eczema	163 (15.9)
Atopy	
Aeroallergen SPT	82 (8.0)
Aeroallergen-specific IgE	618 (64.1)
Helminth parasites	
Any helminth	407 (39.8)
*A. lumbricoides*	265 (25.9)
Intensity	
Light	147 (14.4)
Moderate/heavy	106 (10.4)
*T. trichiura*	78 (7.6)
Intensity	
Light	71 (6.9)
Moderate/heavy	2 (0.2)
*A. duodenale*	10 (1.0)
Intensity	
Light	5 (0.5)
Moderate/heavy	0 (0.0)
*S. stercoralis*	34 (3.3)
*H. nana*	64 (6.3)
*S. haematobium*	23 (2.2)
*E. vermicularis*	11 (1.1)

* In peripheral blood.

**Table 2 ijerph-18-06156-t002:** Univariate analyses for allergic diseases.

Variable	Asthma			Rhinoconjunctivitis			Eczema		
	*N* (%)	OR (95% CI)	*p*	*N* (%)	OR (95% CI)	*p*	*N* (%)	OR (95% CI)	*p*
Sex									
Female	55 (57.9)	1.53 (0.99; 2.33)	0.053	30 (49.2)	1.03 (0.62; 1.74)	0.898	92 (56.4)	1.47 (1.05; 2.06)	0.025
Male	40 (42.1)	1		31 (50.8)	1		71 (43.6)	1	
Age									
5–9	44 (46.3)	1		11 (18.0)	1		67 (41.1)	1	
10–14	51 (53.7)	0.84 (0.55; 1.28)	0.421	50 (82.0)	3.57 (1.83; 6.94)	<0.001	96 (58.9)	1.07 (0.76; 1.50)	0.710
Residence									
Urban	57 (60.0)	0.98 (0.64; 1.50)	0.915	35 (57.4)	0.87 (0.52; 1.47)	0.606	112 (68.7)	1.53 (1.07; 2.19)	0.020
Rural	38 (40.0)	1		26 (42.6)	1		51 (31.3)	1	
Helminths (any)									
No	57 (60.0)	1		47 (77.0)	1		96 (58.9)	1	
Yes	38 (40.0)	1.01 (0.66; 1.55)	0.964	14 (23.0)	0.43 (0.23; 0.79)	0.007	67 (41.1)	1.07 (0.76; 1.50)	0.707
*A. lumbricoides*									
No	67 (71.6)	1		55 (90.2)	1		116 (71.2)	1	
Yes	28 (29.5)	1.22 (0.77; 1.94)	0.405	6 (9.8)	0.30 (0.13; 0.70)	0.005	47 (28.8)	1.19 (0.82; 1.73)	0.352
Intensity			0.411			0.036			0.179
Uninfected	68 (71.6)	1		55 (90.2)	1		117 (71.8)	1	
Light	18 (18.9)	1.44 (0.83; 2.50)	0.195	5 (8.2)	0.46 (1.18; 1.16)	0.101	31 (19.0)	1.49 (0.96; 2.32)	0.077
Moderate/heavy	9 (9.5)	0.96 (0.46; 1.98)	0.908	1 (1.6)	0.12 (0.02; 0.90)	0.039	15 (9.2)	0.92 (0.52; 1.64)	0.778
*T. trichiura*									
No	91 (95.8)	1		60 (98.4)	1		154 (98.5)	1	
Yes	4 (4.2)	0.51 (0.18; 1.42)	0.196	1 (1.6)	0.19 (0.03; 1.49)	0.104	9 (5.5)	0.67 (0.33; 1.37)	0.273
Intensity			0.095			0.312			0.539
Uninfected	91 (95.8)	1		60 (98.4)	1		155 (95.1)	1	
Light	3 (3.2)	0.42 (0.13; 1.35)	0.144	1 (1.6)	0.21 (0.03; 1.55)	0.1277	8 (4.9)	0.65 (0.31; 1.39)	0.266
Moderate/heavy	1 (1.1)	9.44 (0.59; 152.19)	0.114	0 (0.0)	-	-	0 (0.0)	-	-
*A. duodenale*									
No	61 (100.0)			61 (100.0)			161 (98.8)	1	
Yes	0 (0.0)	-	-	0 (0.0)	-	-	2 (1.2)	1.32 (0.28; 6.29)	0.725
*S. stercoralis*									
No	92 (96.8)	1		59 (96.7)	1		160 (98.2)	1	
Yes	3 (3.2)	0.94 (0.28; 3.15)	0.925	2 (3.3)	0.99 (0.23; 4.21)	0.984	3 (1.8)	0.50 (0.15; 1.66)	0.258
*H. nana*									
No	88 (92.6)	1		57 (93.4)	1		152 (93.3)	1	
Yes	7 (7.4)	1.22 (0.54; 2.75)	0.639	4 (6.6)	1.06 (0.37; 3.01)	0.920	11 (6.7)	1.10 (0.52; 2.16)	0.777
*S. haematobium*									
No	95 (100.0)			59 (96.7)	1		157 (96.3)	1	
Yes	0 (0.0)	-	-	2 (3.3)	1.52 (0.35; 6.63)	0.578	6 (3.7)	1.90 (0.74; 4.88)	0.185
*E. vermicularis*									
No	95 (100.0)	-	-	61 (100.0)	-	-	163 (100.0)	-	-
Yes	0 (0.0)			0 (0.0)			0 (0.0)		

**Table 3 ijerph-18-06156-t003:** Multivariate analyses for allergic diseases.

Variable	Asthma		Rhinoconjunctivitis		Eczema	
	OR (95% CI)	*p*	OR (95% CI)	*p*	OR (95% CI)	*p*
Sex						
Female	1.54 (1.00; 2.36)	0.050	0.96 (0.57; 1.63)	0.888	1.47 (1.05; 2.07)	0.026
Male	1		1		1	
Age						
5–9	1		1		1	
10–14	0.84 (0.55; 1.29)	0.433	3.29 (1.68; 6.43)	0.001	1.05 (0.74; 1.49)	0.773
Residence						
Urban	0.96 (0.62; 1.48)	0.856	0.81 (0.47; 1.39)	0.440	1.51 (1.05; 2.18)	0.028
Rural	1		1		1	
*A. lumbricoides*						
No	1		1		1	
Yes	1.26 (0.78; 2.02)	0.344	0.36 (0.15; 0.85)	0.020	1.30 (0.89; 1.91)	0.177
*T. trichiura*						
No	1		1		1	
Yes	0.48 (0.17; 1.35)	0.162	0.21 (0.03; 1.58)	0.131	0.67 (0.33; 1.39)	0.284
*A. duodenale*						
No					1	
Yes	-	-	-	-	1.68 (0.34; 8.24)	0.524
*S. stercoralis*						
No	1		1		1	
Yes	0.85 (0.25; 2.85)	0.788	1.03 (0.23; 4.50)	0.973	0.49 (0.15; 1.64)	0.245
*H. nana*						
No	1		1		1	
Yes	1.28 (0.56; 2.91)	0.561	0.96 (0.33; 2.79)	0.947	1.13 (0.57; 2.24)	0.720
*S. haematobium*						
No			1		1	
Yes	-	-	1.16 (0.26; 5.21)	0.848	1.67 (0.64; 4.38)	0.298

**Table 4 ijerph-18-06156-t004:** Univariate and multivariable analyses for atopy.

Variable	SPT					Aeroallergen-specific IgE				
	*N* (%)	Univariate		Multivariable		*N* (%)	Univariate		Multivariable	
		OR (95% CI)	*p*	OR (95% CI)	*p*		OR (95% CI)	*p*	OR (95% CI)	*p*
Sex										
Female	35 (42.7)	1		1		275 (44.5)	1		1	
Male	47 (57.3)	1.28 (0.81; 2.03)	0.282	1.30 (0.82; 2.05)	0.265	343 (55.5)	1.48 (1.14; 1.93)	0.003	1.51 (1.16; 1.98)	0.003
Age										
5–9	30 (36.6)	1		1		260 (42.1)	1		1	
10–14	52 (63.4)	1.30 (0.82; 2.08)	0.266	1.30 (0.81; 2.09)	0.280	358 (57.9)	1.04 (0.80; 1.36)	0.765	1.13 (0.86; 1.48)	0.400
Residence										
Urban	44 (53.7)	1		1		340 (55.0)	1		1	
Rural	38 (40.0)	1.36 (0.86; 2.14)	0.187	1.39 (0.87; 2.22)	0.163	278 (45.0)	1.83 (1.38; 2.41)	<0.001	1.80 (1.36; 2.40)	<0.001
Helminths (any)										
No	47 (60.0)	1				358 (57.9)	1			
Yes	35 (40.0)	1.14 (0.72; 1.80)	0.576	-	.	260 (42.1)	1.46 (0.96; 1.64)	0.103	-	-
*A. lumbricoides*										
No	60 (73.2)	1		1		440 (71.2)	1		1	
Yes	22 (26.8)	1.05 (0.63; 1.75)	0.842	1.10 (0.65; 1.85)	0.725	178 (28.8)	1.46 (1.07; 1.99)	0.016	1.42 (1–03; 2.40)	0.034
Intensity			0.842					0.026		
Uninfected	60 (73.2)	1				446 (72.2)	1			-
Light	12 (14.6)	1.05 (0.55; 2.01)	0.878	-	-	98 (15.9)	1.52 (1.02; 2.26)	0.038	-	-
Moderate/heavy	10 (12.2)	1.23 (0.61; 2.49)	0.560	-	-	74 (12.0)	1.59 (1.01; 2.50)	0.047	-	-
*T. trichiura*										
No	75 (91.5)	1		1	-	572 (92.6)	1		1	
Yes	7 (8.5)	1.14 (0.51; 2.58)	0.746	1.05 (0.46; 2.40)	0.911	46 (7.4)	0.95 (0.58; 1.56)	0.839	0.81 (0.49; 1.35)	0.417
Intensity			0.991					0.912		
Uninfected	76 (92.7)	1			-	576 (93.2)	1		-	-
Light	6 (7.3)	1.06 (0.45; 2.53)	0.893	-	-	42 (6.8)	0.90 (0.54; 1.49)	0.667	-	-
Moderate/heavy	0 (0.0)	-	-	-	-	0 (0.0)	-	-	-	-
*A. duodenale*										
No	81 (98.8)	1		1		610 (98.7)	1		1	
Yes	1 (1.2)	1.28 (0.16; 10.22)	0.817	1.02 (0.12; 8.42)	0.988	8 (1.3)	2.26 (0.48; 10.68)	0.305	1.63 (0.34; 7.88)	0.541
*S. stercoralis*										
No	82 (100.0)					601 (97.2)	1		1	
Yes	0 (0.0)	-	-	-		17 (2.8)	0.55 (0.28; 1.09)	0.085	0.52 (0.26; 1.05)	0.070
*H. nana*										
No	74 (90.2)	1		1		582 (94.2)	1		1	
Yes	8 (9.8)	1.71 (0.79; 3.72)	0.177	1.72 (0.78; 3.78)	0.179	36 (5.8)	0.91 (0.53; 1.58)	0.739	1.03 (0.58; 1.81)	0.932
*S. haematobium*										
No	79 (96.3)	1		1		607 (98.2)	1		1	
Yes	3 (3.7)	1.75 (0.51; 6.01)	0.375	1.87 (0.53; 6.62)	0.329	11 (1.8)	0.55 (0.24; 1.29)	0.169	0.68 (0.29; 1.62)	0.389
*E. vermicularis*										
No	81 (98.8)	1		1		609 (98.5)	1		1	
Yes	1 (1.2)	1.15 (0.15; 9.09)	0.895	1.24 (0.15; 9.89)	0.842	9 (1.5)	5.10 (0.64; 40.41)	0.123	5.48 (0.68; 44.38)	0.111

## Data Availability

The data presented in this study are available on request from the corresponding author. The data will be made publicly available in a public database as soon as feasible.

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
