# Peer review of "Lack of Consistent Association between Asthma, Allergic Diseases, and Intestinal Helminth Infection in School-Aged Children in the Province of Bengo, Angola"

_ijerph, 2021, doi:10.3390/ijerph18116156_

Round 1

Reviewer 1 Report

Lack of consistent association between asthma, allergic diseases and intestinal helminth infection in school-aged children in the province of Bengo, Angola

Margarete Arrais, Ofélia Lulua, Francisca Quifica, José Rosado-Pinto, Jorge M.R. Gama, Philip J. Cooper, Luis Taborda-Barata,*, Miguel Brito.

This is a very interesting study regarding the controversial correlation between helminth infection and allergic diseases in Angola. The manuscript is well written, easy to follow and well discussed, but some minor questions arise:

  • The authors have measured serum levels of IgE but have not quantified levels of IgG, which have been shown to be of importance to explain this correlation (https://doi.org/10.1111/cea.13055). It would be interesting to include this aspect in their discussion.
  • Another important factor that has been superficially discussed is the length of their infection. In other words, is there any correlation between the time some of these children have been infected with helminths and the parameters measured? Are these new infections? Have these children been infected during infancy? I understand it may be difficult to ascertain for how long they have been infected but previous infections maybe easier to establish. it is not clear if this points were even addressed in their questionnaire. Please, comment
  • Similarly, I understand nearly 40% of these children have been treated against these parasites. It would be interesting to analyse whether this treatment has any impact on the parameters measured in comparison to untreated children. There is some experimental evidence suggesting that some anti-parasite treatments can negatively affect the development of allergic responses and this is my personal observation as well (https://doi.org/10.1038/icb.2009.47).

Author Response

Reviewer #1 Comments

Reviewer comment: Lack of consistent association between asthma, allergic diseases and intestinal helminth infection in school-aged children in the province of Bengo, Angola

Margarete Arrais, Ofélia Lulua, Francisca Quifica, José Rosado-Pinto, Jorge M.R. Gama, Philip J. Cooper, Luis Taborda-Barata, Miguel Brito.

This is a very interesting study regarding the controversial correlation between helminth infection and allergic diseases in Angola. The manuscript is well written, easy to follow and well discussed, but some minor questions arise:

The authors have measured serum levels of IgE but have not quantified levels of IgG, which have been shown to be of importance to explain this correlation (https://doi.org/10.1111/cea.13055). It would be interesting to include this aspect in their discussion.

Authors reply: We really thank the reviewer for this important comment, which will allow us to clarify this point for him. If the comments regard aeroallergen-specific IgE, we did not measure this parameter since its relevance in the context of diagnosis of allergic sensitization is not fully determined. In fact, we only screened for the presence of aeroallergen-specific IgE levels (Phadiatop) as a measure of atopy, and not specifically for each allergen. If the reviewer comments concern screening for helminth-specific IgG as a measure of helminth exposure, we did not consider doing that because such measures are unhelpful in endemic settings where presence of helminth-specific IgG is universal. Such a measure, which cannot distinguish between current and past exposures, is only useful in settings of low prevalence where detection rates of helminths in stool samples are very low.

Reviewer comment: Another important factor that has been superficially discussed is the length of their infection. In other words, is there any correlation between the time some of these children have been infected with helminths and the parameters measured? Are these new infections? Have these children been infected during infancy? I understand it may be difficult to ascertain for how long they have been infected but previous infections maybe easier to establish. it is not clear if this points were even addressed in their questionnaire. Please, comment

Authors reply: Active infections were detected by microscopy of stool samples in this cross-sectional analysis. The only way to establish the presence of infections in infancy would be to do a prospective study from birth or early infancy. There are no other reliable retrospective measures of infant infections with intestinal helminths – recall in (maternal) questionnaires is extremely unreliable and would only detect expulsion of large parasites such as A. lumbricoides. Intestinal helminth parasites can survive for a period of years and parasite burdens reflect cumulative exposures – the presence of active infections particularly at high parasite burdens thus may correspond to infections lasting years but detection of parasites in stool samples at a single point in time provides no indication of infection duration.

Reviewer comment: Similarly, I understand nearly 40% of these children have been treated against these parasites. It would be interesting to analyse whether this treatment has any impact on the parameters measured in comparison to untreated children. There is some experimental evidence suggesting that some anti-parasite treatments can negatively affect the development of allergic responses and this is my personal observation as well (https://doi.org/10.1038/icb.2009.47).

Authors reply: We thank the reviewer for this important comment. We did analyse whether having been treated was associated with an impact on measured outcomes. However, we did not find such an association, possibly because of our study design, which was cross-sectional and not longitudinal. We have added this point in our discussion (page 10, lines 345-354) and also added, in any case, a new reference (Cai et al, 2009) – new reference 40

Reviewer 2 Report

This is an interesting paper about associations between intestinal helmintic infection and allergic diseases. This theme is of general interest, especially considering 2 important facts: the prevalence of allergic diseases is increasing worldwide and the prevalence of helminthiasis in several countries, mainly in Africa and South America, is still high.

I consider the manuscript suitable for publishing, but I would like to clarify the following points, which I will describe bellow.

1. In Abstract, line 29, Tables 2 and 3, lines 227-28 (Results) and lines 243-44 (Discussion) the authors refer to rhinoconjunctivitis (prevalence of 6%). However, in the Results (lines 182 and 183), the information is quite different: rhinitis / rhinoconjunctivites, with prevalence of 21.6%/6.0%, respectively. After reading this (new) information in the Results, it was not clear to me if the analyzes described in Tables 2 and 3 refer to rhinoconjunctivitis (prevalence of 6%), rhinitis (prevalence of 21.6%), or both, which totally modifies our reasoning and conclusions. Please, clarify these points and describe the data accordingly in all subdivisions of the paper (Abstract, Results, Tables, Discussion etc)

2. Introduction, lines 51 to 54.  At this point, a more detailed explanation of the likely immunological mechanism through which helminth infections can potentially reduce the prevalence of allergic diseases would be of great interest. I suggest a brief description of the mechanism and, if possible, a illustration. 

3. Definitions (Lines 144-5).  If a positive test for sIgE defines atopy, and, as discussed in lines 347-8, such measurement may be overestimating the levels of sIgE in a context of endemic helminthiasis, we can conclude that the prevalence of atopy in the present study is overestimated. Within this context, how do the authors explain the findings described in Table 4 (positive association of A. lumbriboides infection with atopy defined by sIgE)?

4. Table 1. How did the authors interpret the findings of helminthiasis in 39.8% of the studied population, but only 9.3% of eosinophilia in peripheral blood? Wouldn't a higher percentage of eosinophilia be expected? 

A curiosity: were the levels of total IgE evaluated in the population along with the sIgE dosage? If so, would this data be relevant to be described in the paper?

5. Finally, the authors make it clear that the objective of the study was to assess the relationship between the presence of helminthiasis and the prevalence of atopy, asthma and allergic diseases in children living in Angola. Accordingly, in the Discussion, the findings of the present study were compared with similar findings in other African countries (Lines 265-339), including the results of 2 metaanalyzes.

Despite this, I suggest to include in the discussion some data from other countries (such as countries in Latin America, China and East Asia) on the theme of helminthiasis and prevalence of allergic diseases.  Thus, we would broaden the discussion and make it more global, and less regional.  

Author Response

Reviewer #2 Comments

Reviewer comment: This is an interesting paper about associations between intestinal helmintic infection and allergic diseases. This theme is of general interest, especially considering 2 important facts: the prevalence of allergic diseases is increasing worldwide and the prevalence of helminthiasis in several countries, mainly in Africa and South America, is still high.

I consider the manuscript suitable for publishing, but I would like to clarify the following points, which I will describe bellow.

  1. In Abstract, line 29, Tables 2 and 3, lines 227-28 (Results) and lines 243-44 (Discussion) the authors refer to rhinoconjunctivitis (prevalence of 6%). However, in the Results (lines 182 and 183), the information is quite different: rhinitis / rhinoconjunctivites, with prevalence of 21.6%/6.0%, respectively. After reading this (new) information in the Results, it was not clear to me if the analyzes described in Tables 2 and 3 refer to rhinoconjunctivitis (prevalence of 6%), rhinitis (prevalence of 21.6%), or both, which totally modifies our reasoning and conclusions. Please, clarify these points and describe the data accordingly in all subdivisions of the paper (Abstract, Results, Tables, Discussion etc).

Authors reply: We thank the reviewer for this very important comment. All of our results on upper airways manifestations have to do with rhinoconjunctivitis. We have re-checked all mentioned sections and corrected the information in Results, line 186 and homogenized the communication of results in all mentioned sections.

Reviewer comment: 2. Introduction, lines 51 to 54.  At this point, a more detailed explanation of the likely immunological mechanism through which helminth infections can potentially reduce the prevalence of allergic diseases would be of great interest. I suggest a brief description of the mechanism and, if possible, a illustration.

Authors reply: We thank the reviewer for this comment and have added a succinct observation on the main possible mechanisms (Introduction, page 2, lines 52-53). Although we also found the suggestion of adding an illustration quite interesting, we did not add an illustration since we really wanted the scope of the article to focus not on the immunological mechanisms but rather on the epidemiological aspects.

Reviewer comment: 3. Definitions (Lines 144-5).  If a positive test for sIgE defines atopy, and, as discussed in lines 347-8, such measurement may be overestimating the levels of sIgE in a context of endemic helminthiasis, we can conclude that the prevalence of atopy in the present study is overestimated. Within this context, how do the authors explain the findings described in Table 4 (positive association of A. lumbriboides infection with atopy defined by sIgE)?

Authors reply: We thank the reviewer for this very pertinent observation. We defined atopy according to two criteria: a) having positive SPT, and b) having positive screening test for allergy-specific IgE (Phadiatop). We believe that sIgE overestimates atopy through the mechanisms outlined in the discussion (pages 10-11, lines 356-386. Although we reported the association between infection with A. lumbricoides and sIgE (as measured by Phadiatop), we did not discuss the observation in detail because we believe it to be an unreliable measure of atopy in our helminth-endemic study setting and so the observation with respect to atopy risk is difficult to interpret.

Reviewer comment: 4. Table 1. How did the authors interpret the findings of helminthiasis in 39.8% of the studied population, but only 9.3% of eosinophilia in peripheral blood? Wouldn't a higher percentage of eosinophilia be expected?

A curiosity: were the levels of total IgE evaluated in the population along with the sIgE dosage? If so, would this data be relevant to be described in the paper?

Authors reply: We really thank the reviewer for these very important comments Total IgE was not measured due to lack of sufficient peripheral blood. The association between eosinophilia and helminth infections adapted to the human host is modified in circumstances of infection chronicity such that eosinophil levels in such populations may be only moderately raised in a proportion of those infected due to the effects of such infections in modulating eosinophil recruitment and activation. The exception are Toxocara spp. which are zoonotic infections unable to complete its life cycle in the human hosts and which are strongly associated with eosinophilia even in helminth-endemic settings.

Reviewer comment: 5. Finally, the authors make it clear that the objective of the study was to assess the relationship between the presence of helminthiasis and the prevalence of atopy, asthma and allergic diseases in children living in Angola. Accordingly, in the Discussion, the findings of the present study were compared with similar findings in other African countries (Lines 265-339), including the results of 2 metaanalyzes.

Despite this, I suggest to include in the discussion some data from other countries (such as countries in Latin America, China and East Asia) on the theme of helminthiasis and prevalence of allergic diseases.  Thus, we would broaden the discussion and make it more global, and less regional. 

Authors reply: We thank the reviewer for this very important comment and have now better highlighted in the Discussion (Page 10; lines 351-354) that studies included in two relevant meta-analyses (Refs 7 and 8) including data not only from Africa but also from other continents show that observed results are not restricted by geographic region.  
